# Chlorine Modulation Fluorescent Performance of Seaweed-Derived Graphene Quantum Dots for Long-Wavelength Excitation Cell-Imaging Application

**DOI:** 10.3390/molecules26164994

**Published:** 2021-08-18

**Authors:** Weitao Li, Ningjia Jiang, Bin Wu, Yuan Liu, Luoman Zhang, Jianxin He

**Affiliations:** 1Textile and Garment Industry of Research Institute, Zhongyuan University of Technology, Zhengzhou 450007, China; m13470061821@163.com (N.J.); z1543086543@163.com (L.Z.); 2Institute of Nanochemistry and Nanobiology, School of Environmental and Chemical Engineering, Shanghai University, Shanghai 200444, China; 3School of Life Science and Technology, Tongji University, Shanghai 200082, China; 4Anhui Institute of Metrology, Hefei 230051, China; 18801911141@163.com

**Keywords:** graphene quantum dots, 633 nm, Cl-doping

## Abstract

Biological imaging is an essential means of disease diagnosis. However, semiconductor quantum dots that are used in bioimaging applications comprise toxic metal elements that are nonbiodegradable, causing serious environmental problems. Herein, we developed a novel ecofriendly solvothermal method that uses ethanol as a solvent and doping with chlorine atoms to prepare highly fluorescent graphene quantum dots (GQDs) from seaweed. The GQDs doped with chlorine atoms exhibit high-intensity white fluorescence. Thus, their preliminary application in bioimaging has been confirmed. In addition, clear cell imaging could be performed at an excitation wavelength of 633 nm.

## 1. Introduction

Today, humankind is faced with some diseases that are extremely difficult to cure, such as cancer. Bioimaging is a method for observing biological processes that can track cell metabolism [1,2,3,4,5]. Therefore, bioimaging technology can be used for disease detection. Many fluorescent nanomaterials have been developed and used in biomedicine, especially for bioimaging [6,7]. The discovery of semiconductor quantum dots has opened a new chapter in modern fluorescent nanomaterials [8,9]. Compared with traditional organic dyes, the unique advantages of semiconductor quantum dots, such as high quantum yield, excellent photostability, and fluorescence tenability, make them promising materials in bioimaging applications. However, they comprise Cd, Se, Hg, and other toxic elements that are nonbiodegradable, causing environmental hazards and toxicity issues. Therefore, we need to explore other nanomaterials that can overcome these shortcomings and replace traditional semiconductor quantum dots.

Although the history of graphene quantum dots (GQDs) is relatively short [10,11], they occupy an essential position in the field of semiconductor luminescent materials owing to their strong stability, excellent photoluminescence performance, good biocompatibility, and environmental friendliness [12,13,14,15,16,17,18,19]. Furthermore, bioimaging enables an in-depth understanding of cellular diseases, obtains crucial information for cancer diagnosis, enhances the monitoring of treatment processes, and enables the development of new drugs. Therefore, it is necessary to apply GQDs in biological imaging. Currently, most GQDs used in bioimaging show green and yellow fluorescence, which has the problem of spectral overlapping and significant loss to cells, because the excitation light has a short wavelength and high energy [20,21,22]. Moreover, other GQDs with longer wavelengths show red fluorescence and are mostly soluble in organic solvents and insoluble in water [23,24], making them unsuitable for bioimaging. Thus, to overcome these limitations, water-soluble GQDs with longer wavelengths are required. Our previous study achieved a redshift in the spectrum of GQDs by optimizing the chlorine (Cl) atom doping and applying it to nuclear imaging under an excitation wavelength of 633 nm; it was the first report of pure GQDs bioimaging using excitation at such a long wavelength [13]. However, most GQDs prepared using biomass carbon as a precursor exhibit blue fluorescence, thereby considerably limiting their application. Consequently, we aimed at preparing GQDs with long-wavelength fluorescence using biomass carbon and doping the GQDs with Cl atoms to improve their performance in biological imaging.

Thus, we designed a simple and universal solvothermal strategy to manufacture novel biomass carbon-based GQDs. Then, we obtained GQDs with different defects by adjusting the synthesis conditions, such as the reaction temperature and the amount of Cl-doping. Finally, we compared their bioimaging effects and observed that the Cl atom-doped GQDs had fewer defects and could be used for bioimaging under an excitation wavelength of 633 nm.

## 2. Materials and Methods

### 2.1. Synthesis of GQDs and Cl-GQDs

Chemicals were purchased and used directly without any further purification. GQDs and Cl-GQDs were synthesized through a solvothermal method using biomass carbon (seaweed) as the precursor. Seaweed was obtained from the Beibu Gulf in Guangxi, China. Organic solvents such as ethanol and chloroform were purchased from Sinopharm.

From our previous studies [13,20,25,26], we concluded that Cl-doping has a regulatory effect on GQDs. In a typical procedure for synthesizing GQDs, 0.1 g of seaweed and 10 mL of ethanol were dispersed in an ultrasonic bath for 10 min. Then, the solution was transferred to a 25 mL Teflon-lined stainless steel autoclave, heated to 180 °C, and reacted for 12 h. After cooling to room temperature, the product was filtered through a 220 nm microporous membrane to remove insoluble carbon residues, providing the GQDs solution. Cl-GQDs were prepared in a mixed solvent using ethanol and chloroform (volume ratio is 7:3), and the other steps were the same as those for the preparation of GQDs.

### 2.2. Structural Characterization

The instruments used for characterizing the samples were the same as those used in our previous studies [12,13]. The photoluminescence (PL) quantum yields (QYs) of GQDs and Cl-GQDs were determined by comparing the integrated PL intensities and absorbance values using Rhodamine B in ethanol as the reference.

### 2.3. Cell Imaging

For culturing HeLa cells and conducting the cytotoxicity assay, the routine experimental method [12,13] was followed. Briefly, HeLa cells were cultured in DMEM medium (New York, NY, USA). About 100,000 cells were seeded in a dish with a diameter of 40 mm at 37 °C under 5% CO_2_/95% air for 24 h. The solution of GQDs or Cl-GQDs was added to the cells in the culture medium with a final concentration of 5 mg·L^−1^. After 2 h, the cells were washed three times with 2 mL of phosphate-buffered saline buffer and then examined under a laser confocal microscope (Leica TCS SP5) using lasers at wavelengths of 488, 543, and 633 nm.

## 3. Results and Discussion

The morphologies of GQDs and Cl-GQDs are shown in Figure 1. The particle size and thickness of GQDs were tested using a transmission electron microscope (TEM) and an atomic force microscope (AFM), respectively. Figure 1a,b shows the uniform dispersion of GQDs and Cl-GQDs with average particle sizes of 2.01 and 4.05 nm, respectively. Cl-GQDs were larger than GQDs, which indicates that Cl-doping is helpful for the growth of GQDs. Using high-resolution transmission electron microscopy (HRTEM, Figure 1c,d), we obtained the single-crystal structure of GQDs. The regular hexagonal symmetric spot in the Fourier transformation diagram is similar to the benzene ring lattice structure of graphene, indicating their single-crystal structure. Notably, the lattice spacing of GQDs is 0.22 nm, whereas that of Cl-GQDs is 0.24 nm. We obtain the specific lattice spacing value because the Cl-doping increases the interplanar space, indicating that the Cl atoms are doped inside the GQDs [13,27]. The results of Figure 1e,f show that the average thicknesseses of GQDs and Cl-GQDs are 3.47 and 3.95 nm, respectively, which are higher than the thickness of GQDs obtained using the bottom-up method [28,29,30].

The structural characterizations of GQDs and Cl-GQDs are shown in Figure 2 and Appendix A. To analyze the structural differences between GQDs and Cl-GQDs, we performed Fourier transform infrared spectroscopy (FT-IR), X-ray photoelectron spectroscopy (XPS), X-ray powder diffraction (XRD), and Raman. FT-IR is used to characterize the functional groups on the surface of GQDs and Cl-GQDs (Appendix A). The stretching vibration peaks at 3500–3000 cm^−1^ represent –NH_2_ and –OH, which is the reason why GQDs and Cl-GQDs are hydrophilic. In addition, comparing Appendix A, a stretching vibration peak at 756.5 cm^−1^ representing C–Cl in Appendix A proves that Cl atoms are doped in the Cl-GQDs. There are three peaks in the XPS spectrum of GQDs, which correspond to C1s (285.65 eV), N1s (398.70 eV), and O1s (532.91 eV). However, the XPS spectrum of Cl-GQDs has four characteristic peaks. Compared with Appendix A, the fourth peak corresponds to Cl2p (197.79 eV) (Figure 2a), which is consistent with the FT-IR of Cl-GQDs. To further analyze the position of Cl, the high-resolution spectra of Cl2p were analyzed (Figure 2e). There are four peaks of –OCCl (197.2 eV), –OC_6_Cl_4_O (198.7 eV), –C_6_H_4_Cl (200.6 eV), and –CH_2_Cl (202.4 eV) that can be obtained, indicating that the Cl atom in GQDs has four covalent bonds [31,32]. Combined with C–Cl (287.7 eV) in the C1s spectrum, it shows that Cl atoms have been successfully doped in the crystals of Cl-GQDs and exist as functional groups on the surface. The elements ratio of different Cl-doping GQDs (Appendix A) shows that the Cl content increases with the addition of trichloromethane (TCM). When TCM was increased to 30%, the Cl content was 2.38%, which did not change much with the increase in TCM. The comparison of N1s shows that the nitrogen content of the Cl-GQDs is reduced, indicating that the Cl-doping replaces the nitrogen atoms. Generally, the above-mentioned test results confirm that there are several –NH_2_-, –OH-, and –Cl-functional groups on the edges of Cl-GQDs and the high amount of Cl-doping will induce the formation of Cl-GQDs. From XRD spectra (Figure 2f and Appendix A), both GQDs and Cl-GQDs show a broad diffraction peak (002) of 4.44 and 4.51 Å, respectively. This indicates that the (002) interplanar spacing increases with the Cl-doping [27], which is consistent with the principle of increasing the lattice spacing in TEM. However, there are two apparent peaks at 1372.59 and 1572.52 cm^−1^ corresponding to the disordered sp^3^ hybrid carbon (D band) and crystalline sp^2^ hybrid structure (G band) in the Raman spectrum (Appendix A). The Cl-GQDs are endowed with the highest I_G_/I_D_ ratio of 1.32, indicating that the Cl-GQDs show a higher degree of graphitization than GQDs. Interestingly, this demonstrates that the Cl-dopant of Cl-GQDs plays a dominant role in accelerating their graphitization degree.

The optical properties (Figure 3) of GQDs and Cl-GQDs are essential for their practical application. GQDs were prepared by cutting seaweed with different solvents, such as ethanol, acetonitrile, toluene, acetone, and N,N-dimethylformamide (DMF). As shown in Figure 3a, the ultraviolet-visible absorption spectrum of GQDs prepared using ethanol as the solvent exhibits better absorption bands at 310, 412, and 506 nm. The normalized PL spectra of GQDs prepared using ethanol also display the widest halfwidth. Therefore, we chose ethanol as the solvent for preparing GQDs. Combining this observation with the results obtained in our previous study on Cl-doping, we infer that ethanol can improve the graphitization degree and reduce defects [13]. When using TCM as the Cl source, with an increase in TCM, we can easily see that the absorption of visible light gradually increases (Figure 3c). Moreover, when the volume of TCM increases to 30%, the absorption increases to 676 nm; however, when the volume of TCM increases further, the absorption decreases. This shows that ethanol, as a solvent, plays a vital role in the growth of GQDs, and the best volume ratio of TCM is 30%. In addition, the fluorescence spectrum widens and shifts toward red with Cl-doping, reaching roughly 490 nm, and the QY of Cl-GQDs increases to 28% from 16% (GQDs). Finally, when the volume ratio of TCM exceeds 30%, the full width at half maximum (FWHM) becomes narrower and the fluorescence shifts to blue. The white fluorescent Cl-GQDs with a FWHM of 154 nm and a fluorescence peak at 490 nm were prepared under the condition that the volume ratio of ethanol and TCM was 7:3.

The white fluorescent Cl-GQDs with a wide FWHM will have a high performance in the field of bioimaging. Therefore, GQDs and Cl-GQDs were converted into water solvents and dropped into HeLa cells, respectively (Figure 4 and Appendix A). The results showed that both GQDs and Cl-GQDs could be excited at 405, 488, and 543 nm. In particular, Cl-GQDs could be excited at 633 nm to image HeLa cells, demonstrating the improvement and optimization of the optical properties of the Cl-GQDs by Cl-doping. Further comparisons of the fluorescence intensity of GQDs and Cl-GQDs showed that the GQDs and Cl-GQDs have the highest imaging brightness under 488 nm excitation (Appendix A). Finally, we tested the toxicity of GQDs and Cl-GQDs. Appendix A shows that they still have low toxicity at high doses (40 mg·L^−1^). Therefore, Cl-GQDs have various potential applications in future bioimaging and biomedical fields.

## 4. Conclusions

In summary, we have turned seaweed into valuable fluorescent nanomaterials. Through Cl-doping, the graphitization degree of Cl-GQDs is improved and the fluorescence stability is enhanced. In addition, the single optical property and low toxicity of GQDs and Cl-GQDs are beneficial for bioimaging. Therefore, cells can be imaged under a long wavelength of 633 nm excitation. In future studies, we will explore the application of this method to photocatalysis and other fields.

## Figures and Tables

**Figure 1 molecules-26-04994-f001:**
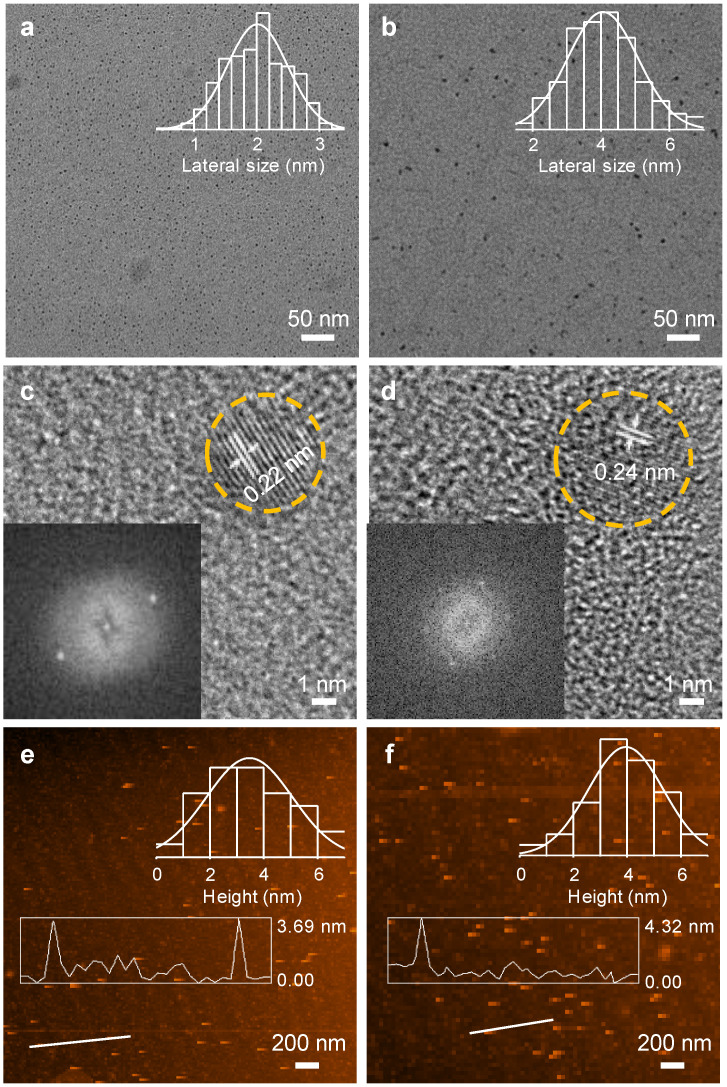
The TEM images and relative size distribution of GQDs (**a**) and Cl-GQDs (**b**). The high-resolution TEM images of GQDs (**c**) and Cl-GQDs (**d**) (insets: fast Fourier transform pattern). AFM images and height distributions of GQDs (**e**) and Cl-GQDs (**f**).

**Figure 2 molecules-26-04994-f002:**
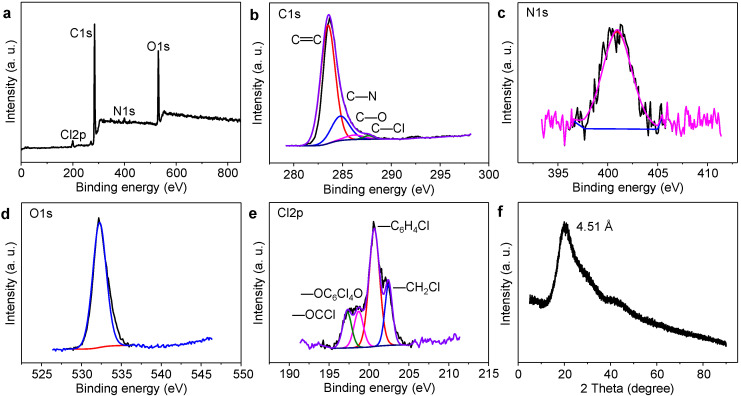
Structure characterization of Cl-GQDs: (**a**) XPS survey spectrum, (**b**) C1s, (**c**) N1s, (**d**) O1s, (**e**) Cl2p, (**f**) XRD patterns of Cl-GQDs.

**Figure 3 molecules-26-04994-f003:**
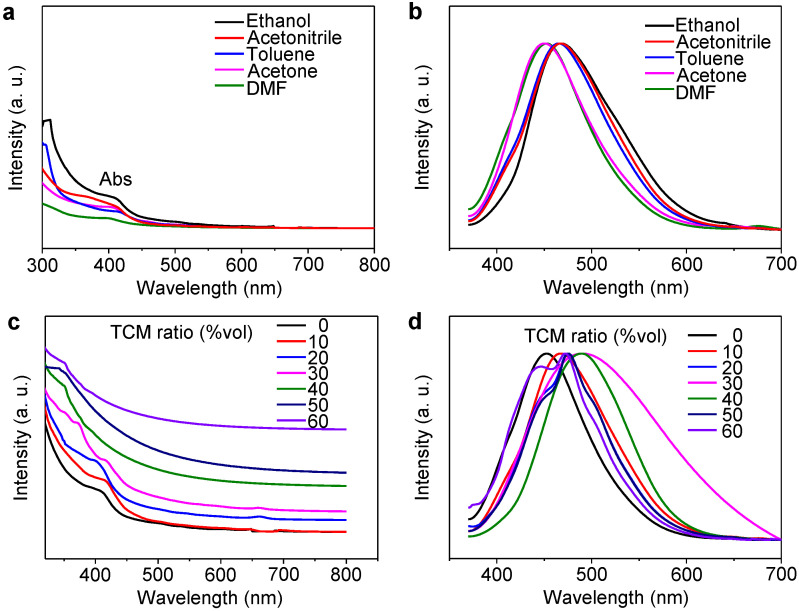
Optical performance of Cl-GQDs and GQDs: (**a**) absorption spectra and (**b**) PL spectra of GQDs in different solvents. (**c**) Absorption and (**d**) PL spectra of different Cl-doped GQDs.

**Figure 4 molecules-26-04994-f004:**
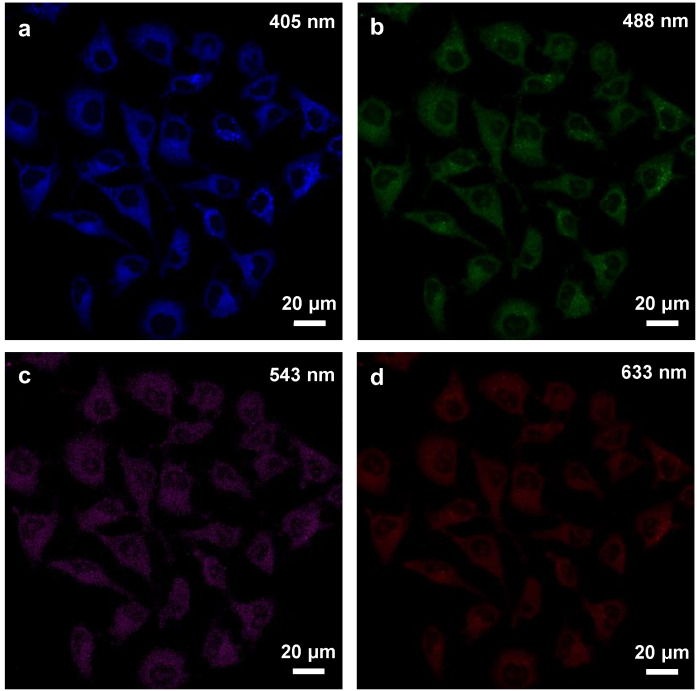
Cell-imaging of Cl-GQDs using HeLa cells excited at (**a**) 405 nm, (**b**) 488 nm, (**c**) 543 nm, and (**d**) 633 nm.

## Data Availability

The data can be made available upon reasonable request.

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
