# Peer review of "Chlorine Modulation Fluorescent Performance of Seaweed-Derived Graphene Quantum Dots for Long-Wavelength Excitation Cell-Imaging Application"

_molecules, 2021, doi:10.3390/molecules26164994_

Round 1

Reviewer 1 Report

In this manuscript, the authors created a novel chlorine doped highly fluorescent graphene quantum dots (Cl-GQDs) from seaweed by using an ecofriendly solvothermal method and proves its application in bioimaging with 633 nm excitation. I recommend a major revision before the paper can be considered for publication due to the following reasons. See below.

1. In Ref 13, the authors also created a Cl doped WGQD using similar method. The authors need to state in detail in the paper about the innovations, technical development etc. of this paper. Basically, what set this paper apart from Ref.13.

2. The authors claimed that Cl-GQDs can be used in cell-imaging with 633 nm excitation. However, as shown in Figure 4 (the authors forgot to label abcd), the image is very dark and blurry. How many Hela cells did the authors use? What is the average signal to noise ratio of the images? Also, the authors should show the image of GQDs with 633 nm excitation as an comparison.

3. The manuscript is poorly written with many grammatical mistakes. I suggest the authors use a professional English editing service and perform an extensive rewriting of the manuscript.

Author Response

In this manuscript, the authors created a novel chlorine doped highly fluorescent graphene quantum dots (Cl-GQDs) from seaweed by using an ecofriendly solvothermal method and proves its application in bioimaging with 633 nm excitation. I recommend a major revision before the paper can be considered for publication due to the following reasons. See below.

In Ref 13, the authors also created a Cl doped WGQD using similar method. The authors need to state in detail in the paper about the innovations, technical development etc. of this paper. Basically, what set this paper apart from Ref.13.

Response: In Ref 13, we used bottom-up method to get GQDs. And in this manuscript, using seaweed as the precursor, GQDs were prepared by top-down method with ethanol and trichloromethane. In the manuscript we emphasized ‘However, most GQDs prepared with biomass carbon as a precursor have blue fluorescence, significantly limiting their application. Consequently, we aimed at preparing long-wavelength fluorescence GQDs based on biomass carbon by doping it with Cl atoms to perform better in biological imaging.’ By preparing seaweed into GQDs, the waste is recycled and reused into valuable GQDs, which is both ecofriendly and energy-saving.

The authors claimed that Cl-GQDs can be used in cell-imaging with 633 nm excitation. However, as shown in Figure 4 (the authors forgot to label abcd), the image is very dark and blurry. How many Hela cells did the authors use? What is the average signal to noise ratio of the images? Also, the authors should show the image of GQDs with 633 nm excitation as an comparison.

Response: Thanks for your advice. Cell-imaging mainly shows that GQDs without Cl-doping could be excited at 405, 488 and 543 nm, but not 633 nm. Particularly, Cl-GQDs could be excited at 633 nm demonstrating the improvement and optimization of the optical property properties of the Cl-GQDs by chlorine Cl-doping. The Figure 4d is not very clear indicating that the Cl-GQDs still need to be improved.

The manuscript is poorly written with many grammatical mistakes. I suggest the authors use a professional English editing service and perform an extensive rewriting of the manuscript.

Response: We have revised seriously the full text.

Reviewer 2 Report

The paper reports on an approach for controlling fluorescence properties of graphene quantum dots using chlorine-doping. The authors upgraded their previously developed protocol for quantum dots production adding an additional step with chlorine doping resulted in a shift of fluorescence emission spectrum to the longer wavelength range. The approach has a potential in biomedical imaging due to the observed shift towards optical transparency window.

However, the biosafety issue is still open. The reported toxicity studies were made on cancer cells culture only (HeLa). The effect on normal cells culture as well as in vivo studies are to be made.  Moreover, the paper misses some control  in two key experiments.

First of all, fluorescence imaging needs indication of the instrumentation empoloyed. Obviously, the reference to earlier paper is not enough. Please, provide the exact details of the employed fluorescence imaging setups and give the control images of cells without quantum dots.

Next, the toxicity studies moved to the supplementary do not show any controls of cells viability without quantum dots applied. Please, add control numbers.

The English requires some corrections. Please, ask a native speaker to reformulate particular sentences and copyedit the paper.

A minor comment:

The 2nd line of the manuscript states that bioimaging is a method for observing biological processes. Actually, bioimaging is a wide class of methods, and their abilities are not limited by monitoring biological processes, they also provide a lot of morphological information.

The paper would be interesting to the audience, however, it requires major revision and another peer-review round.

Author Response

1.First of all, fluorescence imaging needs indication of the instrumentation empoloyed. Obviously, the reference to earlier paper is not enough. Please, provide the exact details of the employed fluorescence imaging setups and give the control images of cells without quantum dots.

Response: The GQDs or Cl-GQDs solution was added to the cells in the culture medium with a final concentration of 5 mg•L-1. 2 hours later, the cells were washing 3 times with 2 mL Pbs buffer and then were examined under a laser confocal microscope (Leica TCS SP5) using lasers 488, 543 and 633 nm. Particularly, Cl-GQDs could be excited at 633 nm to image HeLa cells, which fully demonstrates demonstrating the improvement and optimization of the optical property properties of the Cl-GQDs by chlorine Cl-doping.

The bright field image of HeLa cells was not taken because the bright field image of the confocal was damaged during the test.

Next, the toxicity studies moved to the supplementary do not show any controls of cells viability without quantum dots applied. Please, add control numbers.

Response: Generally, the cell could survival for more than 48 hours without GQDs. And there is no literatures about the control.

The English requires some corrections. Please, ask a native speaker to reformulate particular sentences and copyedit the paper.

Response: We have revised seriously the full text.

Reviewer 3 Report

  1. (Line 72-74) During the preparation of GQDs and Cl-GQDs, are there any treatment steps to remove organic solvents from the system after filtration?
  2. The elemental dope amount of Cl in GQDS should be calculate. The change in the element content of Cl-GQDs prepared by ethanol and chloroform in different ratios should be calculated.
  3. The fluorescence quantum yields of Cl-GQDS and GQDS were compared wih rhodamine. What were the specific values of fluorescence efficiency of Cl-GQDS and GQDS?
  4. Authors mentioned that “Figure S6 shows that they still have low toxicity at high 170 doses (40 mg·L-1). Therefore, Cl-GQDs have huge potential applications in the future bioimaging and biomedical fields.”The toxicity issue need to be resolved before the in vivo application.
  5. In the supplementary literature (Figure S5), why there is no fluorescence image of HeLa cells incubated with GQDS observed at 633 nm?
  6. This Cl-GQDS and GQDS have a wild range of excitation wavelength, so it would be difficult to distinguish the fluorescent signal with other dyes if applied.

Author Response

  1. (Line 72-74) During the preparation of GQDs and Cl-GQDs, are there any treatment steps to remove organic solvents from the system after filtration?

Response: ‘In a typical procedure for synthesing GQDs, 0.1 g of seaweed and 10 ml of ethanol were dispersed in an ultrasonic bath for 10 minutes. Then, the solution was transferred to a 25 ml Teflon-lined stainless steel autoclave, heated to 180 °C, and reacted for 12 h. After cooling to room-temperature, the product was filtered through a 220 nm microporous membrane to remove insoluble carbon products. Finally, the GQDs solution was obtained.’ The process of preparing GQDs don’t remove organic solvents. In this work, only the cell imaging used GQDs and Cl-GQDs removed the organic solvents.

  1. The elemental dope amount of Cl in GQDS should be calculate. The change in the element content of Cl-GQDs prepared by ethanol and chloroform in different ratios should be calculated.

Response: Good suggestion. We have made up the experiment. ‘The elements ratio of different Cl-doping GQDs (Table S1) shows that the content of Cl increases with the added of TCM. When it increases to 30%, the content is 2.38%. And it has not changed much with the increase of TCM.’

Table S1. The elements ratio of different Cl-doping GQDs in XPS survey spectra.

Elements
TCM ratio

Cl (%)

C (%)

O (%)

N (%)

0%

0

76.10

15.96

7.94

10%

0.50

74.00

17.05

8.45

30%

2.38

74.42

19.03

4.17

50%

2.31

74.63

15.56

7.50

  1. The fluorescence quantum yields of Cl-GQDS and GQDS were compared wih rhodamine. What were the specific values of fluorescence efficiency of Cl-GQDS and GQDS?

Response: Good suggestion. The QY of Cl-GQDs increases to 28% from 16% (GQDs).

  1. Authors mentioned that “Figure S6 shows that they still have low toxicity at high 170 doses (40 mg·L-1). Therefore, Cl-GQDs have huge potential applications in the future bioimaging and biomedical fields.” The toxicity issue need to be resolved before the in vivo application.

Response: The concentration of cell-imaging is 5 mg•L-1 which is almost non-toxic. The cell-imaging with GQDs is safe. We will further improve the preparation method of GQDs in the follow-up work, and strive to realize its application in in vivo imaging as soon as possible.

  1. In the supplementary literature (Figure S5), why there is no fluorescence image of HeLa cells incubated with GQDS observed at 633 nm?

Response: Cell-imaging mainly shows that GQDs without Cl-doping could be excited at 405, 488 and 543 nm, but not 633 nm.

  1. This Cl-GQDS and GQDS have a wild range of excitation wavelength, so it would be difficult to distinguish the fluorescent signal with other dyes if applied.

Response: This manuscript mainly report that ‘Through the Cl-doping, the graphitization degree of the Cl-GQDs is improved, and the fluorescence stability is enhanced.’ Later, we will study how to prepare GQDs with a narrow halfwidth, which is more targeted.

Round 2

Reviewer 1 Report

The authors have made some changes to the manuscript, however, they failed to answer most of my scientific questions. See below. 

1. For cell imaging, I asked the authors about the number of cells used and the signal to noise ratios, but the authors did not provide any information. I suggested the authors to show the cell image of GQD with 633 nm excitation as an comparison. However, the authors did not show it.

2. Although the authors claimed that they have revised “seriously” in the main text, it appears that most of their revisions are just minor changes and even some changes are not grammatically correct.

3. Some additional question on Figure 3c, it looks like the intensities of all spectra increase with higher TCM ratios, is this because of an increase of the baseline? If so, the authors should perform a baseline subtraction to show the true intensity of the absorption spectra.

Author Response

1.Response: Thanks for your suggestion. We added the details in the manuscript. ‘Simply, HeLa cells were cultured in DMEM medium (Gibco, USA). About 100,000 cells were seeded in a dish with the diameter of 40 mm at 37 °C under 5% CO2/ 95% air for 24 h. The solution of GQDs or Cl-GQDs was added to the cells in the culture medium with a final concentration of 5 mg•L-1.’

‘Further comparisons of the fluorescence intensity of GQDs and Cl-GQDs showed that the GQDs and Cl-GQDs have the highest imaging brightness under 488 nm excitation (Figure S6).’

  1. Although the authors claimed that they have revised “seriously” in the main text, it appears that most of their revisions are just minor changes and even some changes are not grammatically correct.

Response: We also asked prefessionals to help us review the grammar and revised the full text.

  1. Some additional question on Figure 3c, it looks like the intensities of all spectra increase with higher TCM ratios, is this because of an increase of the baseline? If so, the authors should perform a baseline subtraction to show the true intensity of the absorption spectra.

Response: ‘When using TCM as the Cl source, with an increase in TCM, we can easily see that the absorption of visible gradually increases (Figure 3c). Moreover, when the volume of TCM increases to 30%, the absorption increases to 676 nm, however, when the volume of TCM increases further, the absorption decreases. This shows that ethanol, as a solvent, plays a vital role in the growth of GQDs and the best volume ratio of TCM is 30%.’ The increase of absorption of visible refers to the redshift of the first exciton absorption bands in the direction of longer wavelength, which is consistent with most literature.

Reviewer 2 Report

The author have revised the manuscript, however, the controls are still not shown for both fluorescence imaging (autofluorescence of HeLa cells with quantum dots) and cell viability experiment (the viability for cells that were not incubated with quantum dots).

Moreover, English language still requires editing, since its level was not improved as compared to the previous vesrion of the manuscript.

Author Response

Response: Thanks for the suggestion, we have added the control group to the Figure S7.

Reviewer 3 Report

Acceptable

Author Response

Thank you sincerely for your recognition of this work.